# Basal Forebrain-Dorsal Hippocampus Cholinergic Circuit Regulates Olfactory Associative Learning

**DOI:** 10.3390/ijms23158472

**Published:** 2022-07-30

**Authors:** Yingwei Zheng, Sijue Tao, Yue Liu, Jingjing Liu, Liping Sun, Yawen Zheng, Yu Tian, Peng Su, Xutao Zhu, Fuqiang Xu

**Affiliations:** 1Jiangsu Key Laboratory of Brain Disease and Bioinformation, Research Center for Biochemistry and Molecular Biology, Xuzhou Medical University, Xuzhou 221004, China; northstar1979@163.com (Y.Z.); lunasun1216@126.com (L.S.); 202003010202@stu.xzhmu.edu.cn (Y.Z.); 2State Key Laboratory of Magnetic Resonance and Atomic and Molecular Physics, Key Laboratory of Magnetic Resonance in Biological Systems, Wuhan Center for Magnetic Resonance, Innovation Academy for Precision Measurement Science and Technology, Chinese Academy of Sciences, Wuhan 430071, China; taosj12@lzu.edu.cn (S.T.); liuyue@apm.ac.cn (Y.L.); tttyyy4979@163.com (Y.T.); 3Shenzhen Key Laboratory of Viral Vectors for Biomedicine, Key Laboratory of Quality Control Technology for Virus-Based Therapeutics, Guangdong Provincial Medical Products Administration, NMPA Key Laboratory for Research and Evaluation of Viral Vector Technology in Cell and Gene Therapy Medicinal Products, The Brain Cognition and Brain Disease Institute (BCBDI), Shenzhen Institute of Advanced Technology, Chinese Academy of Sciences, Shenzhen-Hong Kong Institute of Brain Science-Shenzhen Fundamental Research Institutions, Shenzhen 518055, China; jj.liu1@siat.ac.cn (J.L.); hustsupeng@163.com (P.S.); 4University of the Chinese Academy of Sciences, Beijing 100049, China; 5Center for Excellence in Brain Science and Intelligence Technology, Chinese Academy of Sciences, Shanghai 200031, China

**Keywords:** basal forebrain cholinergic neurons, hippocampus, olfactory associative memory, anterograde monosynaptic tracing, rabies virus

## Abstract

The basal forebrain, an anatomically heterogeneous brain area containing multiple distinct subregions and neuronal populations, innervates many brain regions including the hippocampus (HIP), a key brain region responsible for learning and memory. Although recent studies have revealed that basal forebrain cholinergic neurons (BFCNs) are involved in olfactory associative learning and memory, the potential neural circuit is not clearly dissected yet. Here, using an anterograde monosynaptic tracing strategy, we revealed that BFCNs in different subregions projected to many brain areas, but with significant differentiations. Our rabies virus retrograde tracing results found that the dorsal HIP (dHIP) received heavy projections from the cholinergic neurons in the nucleus of the horizontal limb of the diagonal band (HDB), magnocellular preoptic nucleus (MCPO), and substantia innominate (SI) brain regions, which are known as the HMS complex (HMSc). Functionally, fiber photometry showed that cholinergic neurons in the HMSc were significantly activated in odor-cued go/no-go discrimination tasks. Moreover, specific depletion of the HMSc cholinergic neurons innervating the dHIP significantly decreased the performance accuracies in odor-cued go/no-go discrimination tasks. Taken together, these studies provided detailed information about the projections of different BFCN subpopulations and revealed that the HMSc-dHIP cholinergic circuit plays a crucial role in regulating olfactory associative learning.

## 1. Introduction

Accumulating evidence shows that basal forebrain cholinergic neurons (BFCNs) are involved in olfactory associative learning. The basal forebrain (BF), a heterogeneous brain area, consists of multiple subregions, including the medial septum (MS), diagonal band nuclei (DB), substantia innominate (SI), nucleus basalis magnocellularis (nBm), ventral pallidum (VP), extended amygdala, and peripallidal regions. Meanwhile, it contains diverse neuronal populations such as cholinergic neurons, excitatory neurons, and various interneurons. BFCNs, characterized by huge somas with long axons and enormous branches, innervate the neocortex, archeocortex, and other subcortical structures [1]. Recent mapping studies have dissected a portion of BFCN circuits of distinct subregions using anterograde and retrograde tracers [2,3,4,5,6], revealing that different subregions have distinct input and output patterns [7,8]. These anatomical heterogeneities of BFCNs lay the basis for their diverse functions [3,4,9]. BFCNs in the nucleus of the horizontal limb of the diagonal band (HDB) innervate the olfactory bulb (OB) and receive the input directly from the OB and multiple olfactory cortices [4]. They participate in olfactory information processing, arousing, attention, and learning [10,11,12,13,14,15,16]. Real-time monitoring of somas through fiber photometry demonstrates that BFCNs are activated during odor-cued go/no-go tasks [17], whereas c-fos immunofluorescent labeling reveals cholinergic neurons in the HDB, magnocellular preoptic nucleus (MCPO), and SI brain regions (HMS complex, HMSc), which are activated in olfactory associative learning [4]. Moreover, a selective lesion of cholinergic neurons in the MS/vertical limb of the diagonal band (VDB) and the nBm/SI causes different deficits in the social transmission of food preference tasks [9]. Although these findings provided anatomical and functional evidence to support the roles of different subregional BFCNs in olfactory associative learning, the exact cholinergic circuits involved are not elucidated yet.

Olfactory associative learning is associated with multiple brain areas, including the olfactory system and limbic system. The hippocampus (HIP) is one of the crucial brain areas in the limbic system. Like the BF, the HIP is also an anatomically and functionally heterogeneous brain region, which plays an important role in learning, memory, emotion, and cognition [18,19,20,21]. The HIP can be divided into the dorsal and ventral parts (dHIP and vHIP, respectively). Both the dHIP and vHIP can be subdivided further into field CA1 (CA1), field CA2 (CA2), field CA3 (CA3), the dentate gyrus (DG), and the subiculum (SUB). The distribution of neurons in the HIP is arranged in a certain topological pattern. Functionally, the dHIP encodes spatial navigation and formation of declarative memory, whereas the vHIP encodes for emotional behaviors [18,22]. The DG discriminates olfactory information and enhances the learned sensory stimuli in associative learning [23], whereas cell loss in CA1 and the DG disrupts the learning in olfactory associative tasks in adult rats [24]. A lesion of the dorsal DG impairs the integration of odor stimulus and context [25]. These studies revealed that different HIP subregions play diverse roles in olfactory associative learning.

Anatomical and functional evidence supports strong connections between the HIP and BF [4,26]. Preliminary analysis based on retrograde tracing combined with immunochemistry revealed the proportions and distribution patterns of BFNCs in the MS projecting to the HIP [27]. Further studies revealed that the VDB and MS [28] are the major resources of cholinergic projections to the HIP [29], but with significant differences. The CA1 pyramidal cell layers and DG granule cell layers in the dHIP receive inputs almost exclusively from the VDB, whereas those cell layers in the vHIP receive from both the VDB and MS. Our recent study revealed a higher proportional cholinergic projection in the HMSc [4]. Although the connectivity between BFCNs and the dHIP has been preliminarily dissected, the details still need to be further explored and clarified.

Here, we exploited both anterograde monosynaptic and retrograde tracing strategies to analyze the relationships between different subpopulations of BFCNs and the dHIP, and the possible functions involved. The tracing results showed that more than 80% of BFCNs in the HMSc innervated the dHIP. Fiber photometry recording revealed that BFCNs in the HMSc were activated in odor-cued go/no-go discrimination tasks. Further, selected depletion of cells innervating the dHIP impaired the performance of repetitive learning during go/no-go training sessions. Our results identified a critical role of the specific BFCN-dHIP circuit in olfactory associative learning.

## 2. Results

### 2.1. The Output Patterns of BFCNs in Different Subregions

To investigate the whole-brain output of BFCNs, an AAV-HSV-based cell-type-specific monosynaptic anterograde tracing system was used [30]. First, helper viruses (rAAV-UL26.5p-DIO-cmgD-WPRE-pA and rAAV-hSyn-DIO-EGFP-T2A-Her2CT9-pA) were injected into different subregions (MS/DB, HMSc, and nBm) of the BF in ChAT-Cre mice to express the Her2 receptor (Her2CT9) and wild-type HSV gD protein in cholinergic neurons. Three weeks later, gD-null and Her2-targeting HSV-∆gD-hUbc-mcherry-2A-scHer2::gD-WPRE-pA was injected into the same position and specifically infected neurons expressing Her2CT9. After transduction, the HSV genome was transcribed to produce progeny viruses, which spread from the starter neurons to the postsynaptic neurons. Because the gD is not present in the postsynaptic neurons, HSV cannot package into new progeny viruses and spread further (Figure 1A). The starter cells (marked by EGFP and mCherry fluorescence) were restricted to ChAT-positive neurons in the target region (Figure 1B,C). The monosynaptic outputs of BFCNs were labeled by HSV and identified by mCherry only. Imaging of mCherry^+^ neurons showed that cholinergic neurons in the MS/DB directly projected to many brain regions, including the HDB, CA3, CA1, lateral hypothalamic area (LHA), lateral entorhinal cortex (Lent), and supramammillary nucleus (SUM) (Figure 2A). The HMSc cholinergic neurons directly projected to the MS, lateral septal nucleus (LS), LHA, periaqueductal gray (PAG), laterodorsal tegmental nucleus (LDT), CA1, CA2, CA3, and DG (Figure 2B), whereas the nBm cholinergic neurons projected to the somatosensory areas (SS), subparafascicular nucleus (SPF), PAG, parabrachial nucleus (PB), and caudoputamen (CP) (Figure 2C). To reveal the whole-brain efferent patterns of BFCNs, we analyzed the distribution and proportion of mCherry^+^ neurons in different brain regions of the samples from these three injection sites. We only displayed regions with proportions ≥ 1% of the total. The data showed that the mCherry^+^ neurons were mostly (≥5%) found in: the HIP, retrohippocampal region (RHP), LS, LHA, and lateral preoptic area (LPO) for the MS; HIP, LS, medial septal complex (MSC), LHA, and PAG for the HMSc; and somatomotor areas (MO), SS, CP, midbrain reticular nucleus (MRN), and PAG for the nBm (Figure 3A). We further pooled the mCherry^+^ neuron signals of every individual brain region into several intact brain areas according to the Allen Mouse Brain Atlas (2011) (available from http://mouse.brain-map.org/, accessed on 17 March 2022). (http://www.brain-map.org/, accessed on 17 March 2022). We found the preferred targets of BFCNs in different subregions: the MS/DB to the HIP, hypothalamus, and midbrain; the HMSc to the isocortex, HIP, pallidum, thalamus/epithalamus, hypothalamus, midbrain, and hindbrain; and the nBm to the isocortex, striatum, pallidum, thalamus/epithalamus, hypothalamus, and midbrain (Figure 3B). Except for the nBm, the projections of the MS/DB to the HIP and the HMSc to the HIP were mostly spread between the bregma at −2.00~−3.00 mm, which are primary regions of the dHIP (Appendix A). Together, these results revealed that BFCNs in different subregions have distinct output patterns and variable projection relationships with the HIP. 

### 2.2. Topography of BFCN Projecting to Dorsal Hippocampus

To map the projection of the BFCN to the dHIP, we injected RV-ΔG-mCherry or RV-ΔG-EGFP separately into two randomly selected dHIP subregions (CA1, CA3, and DG) in the right hemisphere of the mouse brain (Figure 4A). The BF along the anterior–posterior (AP) axis was divided into serial coronal sections (Figure 4B). Cholinergic neurons were identified by immunofluorescence staining with an anti-ChAT antibody, and then, the fluorescent co-labelled were quantified. It is well known that there are three types of neurons in the BF: cholinergic, GABAergic, and glutamatergic neurons. For the three injection combinations of DG/CA1, CA1/CA3, and DG/CA3, of all labeled cells, the percentage of the cholinergic neurons innervating either region and both regions were 81.701% and 13.273% (Figure 4C), 80.845% and 17.641% (Figure 4D), and 58.957% and 19.344% (Figure 4E), respectively. Moreover, the MS/DB had more labeled neurons projecting to the dHIP than the other two subregions (Figure 4F–H). Considering the few numbers of cholinergic neurons in the nBm projecting to the hippocampus, we only compared the labeled cholinergic neurons in the MS/DB and HMSc. We found that the proportion of labeled cholinergic neurons in the HMSc was significantly higher than in the MS/DB (Appendix A). Generally, 73.6% and 83.5% of dHIP-projecting neurons were cholinergic in the MS/DB and HMSc, respectively (Appendix A). Among all the three dHIP subregions, distribution patterns of labeled cholinergic neurons were similar. The HMSc contributed the most in CA1 (CA1: HMSc vs. MS/DB, *p* = 0.01, Appendix A). In conclusion, our results revealed that BFCNs projecting to the dHIP were topographically organized, and a higher proportion of cholinergic neurons projecting to the dHIP were localized in the HMSc. 

### 2.3. BFCNs in HMSc Were Activated in Odor-Cued Go/No-Go Discrimination Task

Several studies demonstrated that the cholinergic neurons in the HDB/MCPO participated in olfactory associative memory [16]. The finding was consistent with our previous study, which demonstrated by c-fos immunofluorescence staining that cholinergic neurons in the HMSc were activated in odor-cued go/no-go tasks [4].To directly monitor the population activities of cholinergic neurons in the HMSc in the olfactory associative memory behavioral paradigm, we injected the Cre-dependent, adeno-associated virus rAAV-EF1α-DIO-GCaMP6s-WPRE-hGH-pA into the left HMSc and implanted chronically implantable optic fibers in the ipsilateral HMSc in ChAT-Cre mice with fiber-optic cannulae positioned above the viral injection site (Figure 5A,B). Three weeks after viral injection, the Ca^2+^ indicator GCaMP6 was widely expressed in 91.5% of all cholinergic neurons in the HMSc (Figure 5C). Viral expressions and recording targets of optic fibers were verified post hoc in all mice via histological sections and immunofluorescence staining (Figure 5C).

Before the go/no-go task, the mice in GCaMP6 and control groups were trained to lick water. To identify whether the changes of Ca^2+^ signals derived from the action of licking water, the deprived-water mice were recorded using fiber photometry during free water-licking. We found that cholinergic neurons in neither of the two groups showed any changes in fluorescent signals during water-licking actions (Appendix A). 

During go/no-go training, the mice received 100 trials per session repetitively. In each trial, a 2 s time window of odor delivery was followed by a 2 s time window of water delivery. For the odor-cued discrimination task, mice were trained to differentiate two odorants, isoamyl acetate as a positive conditioned stimulus (CS+) with delayed water delivery as a reward, and 2-heptanone as a negative conditioned stimulus (CS−) without water delivery as a no-reward. The two odorants were presented randomly in each trial. All experiments started at the same time for three continuous days. During go/no-go training, we evaluated the responses of the cholinergic population for odor coupled with reward or no-reward in head-fixed mice. For the twenty-five responsive records from five mice, the average signal peaks (ΔF/F) were calculated. We found that cholinergic neurons with GCaMP6 responded to both S+ and S− odor signals (Figure 5D–I), but their responses were not constant during stimulation and reward delivery in the go/no-go task. Specifically, during the odor delivery phases, the Ca^2+^ signals sharply increased and peaked as soon as the odor cues emerged during the learning stage for both CS+ and CS−, whereas the cue-evoked Ca^2+^ signals gradually declined through the water delivery period. The responses for the S+ were stronger than those for S− signals, either during the initial stage of learning or after learning (in initial stage: ΔF/F for S+ = 0.27 ± 0.28%, ΔF/F for S− = 0.22 ± 0.49%; in final stage: ΔF/F for S+ = 0.14 ± 0.22%, ΔF/F for S− = 0.12 ± 0.40%; all *p*-values < 0.05 for ΔF/F for S+ vs. S− in both initial and final stage, Figure 5D–I). The response latencies for S+ and S− signals were gradually reduced along the training process (Figure 5D–I). The cholinergic neurons, expressing only EGFP, did not display any changes in fluorescent signals during the training session (Appendix A). In short, these results revealed that the cholinergic neuronal population in the HMSc was activated in olfactory associative learning.

### 2.4. Selective Depletion of Cholinergic Neurons Innervating dHIP Impaired Correct Performance of Odor-Cued Go/No-Go Discrimination Task

Since BFCNs in the HMSc had a strong projection to the HIP and were activated in the odor-cued discrimination task, we hypothesized that BFCNs in the HMSc innervating the dHIP were involved in olfactory associative learning. In RV tracing, the results showed that more than 80% of all labeled neurons were cholinergic neurons projected to CA1 and/or the DG. Thus, we utilized rAAV-Retro-hSyn-SV40-NLS-Cre-WPRE-hGH-pA with rAAV-DIO-taCasp3-T2A-TEVp-WPRE-hGH-pA to selectively kill cholinergic neurons projecting to the dHIP, and explore their effects on olfaction and olfactory associative memory of mice (Figure 6A). The killing efficiency of the virus was evaluated by counting the number of cholinergic neurons in the HMSc. The number of cholinergic neurons in the Casp3 groups was significantly less than that in the control groups (control group: 14.88 ± 1.36; Casp3 group: 8.75 ± 0.76, *p* < 0.001, Figure 6M and Appendix A). In the buried food test, we found that odor perception ability was not significantly different between the control and Casp3 groups, no matter whether the food was visible or not (Figure 6B–D). Meanwhile, the odor discrimination test was also carried out between the two groups (Figure 6E). Similarly, the mice in both groups showed intact odor discrimination abilities (Figure 6F–H), although the discrimination index was mildly decreased in the Casp3 group, but with no significant difference between the two groups (Figure 6H). These results above demonstrated that selectively killing cholinergic neurons projecting to the dHIP did not affect the odor detection and discrimination abilities of the mice. 

The selective depletion effect on learning and memory was investigated by training free-moving mice on a go/no-go paradigm (Figure 6I,J, see Materials and Methods). During the go/go stage, both groups were trained to learn to lick water within the time window of two odors coupled with water delivery. When water-licking accuracy reached 90% or higher, the mice were used for subsequent go/no-go training. In 3 consecutive days of repeated training, we found there was a similar pattern in correct responses for the S+ signal between the two groups (Figure 6K), because nothing needs to be learned. However, the learning curve for the S-signal was interesting. As expected, the correct rate for the control group improved with training. In comparison, the rate in the Casp3 group was significantly lower in all three days, and there was much less improvement with training (Figure 6L). Taken together, these results revealed that selective depletion of cholinergic neurons impaired the performance of odor-cued go/no-go tasks. 

## 3. Discussion

In this study, we mapped the projectomes of BFCNs and delineated the connection relationships of BFCNs in different subregions and different dHIP subregions. More importantly, we clarified the role of the BF-dHIP cholinergic circuit in olfactory associative learning. 

### 3.1. Projectomes of BFCNs

Via the monosynaptic anterograde tracing system, we dissected the output patterns of BFCNs in different subregions at the whole-brain level. We found that BFCNs in different subregions had distinct efferent patterns. For cholinergic neurons in the MS/DB, they provided most cholinergic innervation to the hippocampal formation (HPF) and hypothalamus, followed by the midbrain (Figure 3A,B). Cholinergic neurons in the nBm mainly projected to the isocortex and midbrain (Figure 3A,B), whereas those within the HMSc had a diffuse projection pattern of cholinergic fibers distributed over multiple brain regions, such as the hypothalamus, HPF, isocortex, pallidum, and midbrain (Figure 3A,B). In addition, compared to the MS/DB and nBm, more cholinergic neurons in the HMSc predominantly innervated olfactory areas and the hindbrain (Figure 3A,B). An array of studies provided anatomical evidence supporting that BFCNs innervate multiple brain areas [3,6]. Consistently, previous studies demonstrated that the MS/DB constructed the septal-hippocampal pathway and provided predominant cholinergic fiber to the HIP [31,32]. Additionally, cholinergic neurons in the nBm projected to the neocortex and amygdala [31,32]. Those in the HDB, one brain area of the HMSc, innervated the olfactory bulb and olfactory cortices [31,32]. However, our study systematically revealed the distribution of the projection brain region and projection intensity of BFCNs in different subregions at the whole-brain level.

### 3.2. Anatomical Relationship between BFCNs and dHIP

Accumulating evidence shows that BFCNs have strong connections with the dHIP [4,26]. Our study demonstrated that the MS/DB provided predominant cholinergic innervations to the dHIP, which is consistent with previous studies describing the afferent of the septo-hippocampal circuit (Figure 3A and Figure 4F–H). Intriguingly, we also found that a higher proportion of cholinergic neurons in the HMSc projected to the dHIP than that in the MS/DB, although the total number of neurons projecting to the dHIP in the HMSc was less than that in the MS/DB (Figure 3B and Figure 4F–H). In our previous studies, we demonstrated that cholinergic neurons in the HMSc are activated in olfactory associative memory [4]. In the present study, we found that calcium signals in the BF cholinergic neuronal population increased for both S+ and S− signals in the odor-cued go/no-go task (Figure 5D–I). Moreover, calcium signals of BFCNs were transiently elevated as soon as odorant stimuli appeared in this behavioral paradigm, whether the reward was coupled or not (Figure 5D–I). Along with the increased accuracies of learning performance, cholinergic neurons’ responses for reward or no-reward coupling with odorant stimuli appeared progressively earlier (Figure 5D–I). However, the cholinergic neuronal population showed no responses for only licking water behavior (Appendix A). These results implied that BFCNs participated in integrating anticipation and learning behavior in the odor-cued go/no-go discrimination tasks, but not only licking behavior. Importantly, these results highly agreed with earlier studies [14]. Several studies demonstrated that cholinergic neurons in the HDB are involved in odor perception and olfactory associative learning [13,16]. A recent electrophysiological study based on highly sensitive tetrode recordings demonstrated that cholinergic neurons participated in the anticipatory activity, sensory discrimination, and reward responses in the odor-cued go/no-go behavioral paradigm [14]. Additionally, through fiber photometry combined with genetically encoded acetylcholine indicator GAChR2.0, acetylcholine levels within the basal forebrain rapidly elevated during reward-seeking behaviors, and were strongly suppressed by reward delivery in odor-cued go/no-go discrimination tasks [17]. However, the specific cholinergic circuit in the HMSc involved in olfactory associative memory was still unclear.

### 3.3. Functions of HMSc-dHIP Circuit in Olfactory Associative Learning

Many studies demonstrated that BFCNs play a crucial role in attention, motivation, sensory cue detection, and learning and memory [12,15]. Correspondingly, substantial studies revealed that different cholinergic circuits drive distinct biological functions [9,33,34,35]. Numerous pharmacological experiments demonstrated that the cholinergic corticopetal circuit regulates attention, motivation, and sensory cue detection, whereas BFCNs projecting to the HIP take charge of learning and memory [36,37]. An odor-cued go/no-go task is an associative learning and memory task, which involves anticipation, motivation, choice-making, learning, and memory. Therefore, we might speculate that go/no-go behavior is likely to be regulated by multiple neuronal circuits. 

In this study, we found that selectively killing the neurons projecting to the dHIP did not affect odor detection and discrimination abilities of mice, but impaired the learning for odor-cued go/no-go tasks (Figure 6). For the S+ signal test, the responses of mice in the two groups showed no difference; however, the accuracies of responses to the S-signal were significantly lower in the selectively neuron-killing mice (Figure 6L,M). These results suggested that selective deletion of the cholinergic neuronal population projecting to the hippocampus did not affect their functions of regulating odor perception and anticipation, but impaired the learning and memory of odor-coupled rewards. Furthermore, our results showed that selective depletion of cholinergic neurons innervating the dHIP does not affect the learning of the go/go task, but significantly affects the ability to correctly reject the no-reward in the go/no-go task (Figure 6). One of the interpretations is that cholinergic neurons are involved only in relearning. Another interpretation is that they are involved in more general odor associative relearning. However, for a simple odor associative learning such as a go/go paradigm, the remaining 60% after selectively depletion might be enough, whereas for a complex paradigm such as go/no-go, the remaining cholinergic neurons in the HMSc are not sufficient.

A drawback of this paper is that we assumed that all neurons in the HMSc brain region projecting to the dHIP were cholinergic neurons. However, our RV tracing results demonstrated that nearly 85% of the projection neurons in the HMSc are cholinergic, and the rest are other types of neurons (Figure 4C–E). Therefore, we hypothesized that cholinergic neurons were responsible for the impairment in the odor-cued go/no-go task mediated by selectively killing neurons projecting to the dHIP, whereas the other types of neurons were not excluded. To further clarify the specific role of cholinergic neurons in the BF-HIP circuit in olfactory associative memory, experiments of specially killing or inhibition of cholinergic neurons in the HMSc projecting to the dHIP need to be conducted.

In summary, we mapped out the output networks of three different subregions of the BF and clarified the anatomical connection relationships between cholinergic neurons in the BF and dHIP. Furthermore, the role of the basal forebrain-hippocampal cholinergic circuit in olfactory associative learning was demonstrated as participating in the learning and memory of odor-cued go/no-go tasks, but not in the perception of odor signals nor licking behavioral anticipation. These results provided the potential circuit mechanisms for how the BFCNs in the HMSc were involved in olfactory associative learning.

## 4. Materials and Methods

### 4.1. Animal

All animal procedures were approved by the Animal Care and Use Committee (Innovation Academy for Precision Measurement Science and Technology, Chinese Academy of Sciences; and Xuzhou Medical University). ChAT-IRES-Cre mice (hereafter referred to as ChAT-Cre) were gifts from Professor Cheng Xiao (Jackson, Strain #006410), and used for anterograde monosynaptic tracing and fiber photometry. Male C57BL/6 mice (Beijing Vital River Laboratory Animal Technology Co., Ltd., Beijing, China) were used for retrograde virus tracing and the Casp3 depletion test. All mice used in this study were 8–12 weeks old. They were bred and housed on a 12/12 light/dark cycle with food and water *ad libitum*. All tests were conducted in the light phase. 

### 4.2. Stereotaxic Injection of the Viruses 

For virus injection, mice were anesthetized with sodium pentobarbital (80 mg/kg) and placed on a stereotaxic apparatus (RWD, 68030, Shenzhen, China). The craniotomy was made with a dental drill (STRONG, Guangdong, China) and removed carefully with a curved needle. The desired viruses were infused into the targeted brain regions using a glass micropipette connected to a microinjector (WPI, 4878, Sarasota, FL, USA), which was driven by a syringe pump (Stoelting, Quintessential Stereotaxic Injector, 53311, Wood Dale, IL, USA). After injection, the micropipette was left in place for 10 min before being slowly withdrawn. The animals were placed on a heating pad to regain consciousness. 

For antegrade tracing, the mixture of rAAV-UL26.5p-DIO-cmgD-WPRE-pA and rAAV-hSyn-DIO-EGFP-T2A-Her2CT9-pA (volume ratio: 8:2, 100 nL in total) was injected into the targeted brain regions. Injection sites of subregions were listed as follows: MS/DB: A-P, 1.00 mm; M-L, −2.65 mm; D-V, −4.60 mm, 14° angle from the horizontal axis; HMSc: A-P, 0.50 mm; M-L, −0.90 mm; D-V, −5.30 mm; nBm: A-P, −0.70 mm; M-L, −2.00 mm; D-V, −4.30 mm. After three weeks, 200 nL of HSV-∆gD-hUbc-mcherry-2A-scHer2::gD-WPRE-pA was injected into the same position following the same injection protocol. Five days later, the mice were euthanized for histological studies. The viruses used in the antegrade tracing experiment were constructed as previously reported [9,30]. A total of 11 male ChAT-Cre mice aged 8–9 weeks were used; 4, 3, and 4 for the MS/DB, HMSc, and nBm group, respectively.

For RV tracing, the same viral injection procedure was performed as described above. Injection sites in the dHIP were: CA1 (A-P, −1.80 mm; M-L, −1.40mm; D-V, −1.55mm), CA3 (A-P, −1.70 mm; M-L, −2.00 mm; D-V, −2.00 mm), and DG (A-P, −1.70mm; M-L, −0.90 mm; D-V, −2.00 mm). In each mouse, RV-ΔG-mCherry and RV-ΔG-EGFP were injected separately into two randomly selected subregions of the dHIP. Five days later, the mice were euthanized for histological studies. Nine male C57 mice aged 8–9 weeks were used in the study.

For fiber photometry recording, 200 nL of rAAV-EF1α-DIO-GCaMP6s-WPRE-hGH-pA or rAAV-EF1α-DIO-eYFP-WPRE-hGH-pA was injected into the unilateral HMSc of the ChAT-Cre mice. The mice injected with rAAV-EF1α-DIO-GCaMP6s-WPRE-hGH-pA were set as the GCaMP6s group and the mice injected with rAAV-EF1α-DIO-eYFP-WPRE-hGH-pA were set as the control group. The injection sites of the HMSc are previously described. Two weeks later, the mice were used for fiber photometry recording. Ten male ChAT-Cre mice at 8–10 weeks of age were used for the following go/no-go testing. 

For the Casp3 depletion study, the rAAV-Retro-hSyn-SV40 NLS-Cre-WPRE-hGH-pA virus was injected into the bilateral dHIP (A-P, −1.70 mm; M-L, ±1.10 mm; D-V, −1.65 mm). Two weeks later, the mice received AAV-EF1α-DIO-taCasp3-T2A-TEVp-WPRE-Hgh-pA or rAAV-EF1α-DIO-taCasp3-EGFP-WPRE-hGH-pA in the bilateral HMSc. The mice injected with AAV-EF1α-DIO-taCasp3-T2A-TEVp-WPRE-Hgh-pA were set as the Casp3 group and the mice injected with rAAV-EF1α-DIO-taCasp3-EGFP-WPRE-hGH-pA were set as the control group. The injection sites of the HMSc are previously described. Two weeks later, the mice were used for the following behavioral testing. In total, 17 male C57 mice aged 8–9 weeks were used in the study with 7 and 10 mice in the control and Casp3 groups, respectively. After the experiment, the mice were used for histological studies. The viral information used in this study is listed in Table 1.

### 4.3. Histology and Immunofluorescence Staining

Mice were anesthetized with an overdose of sodium pentobarbital (200 mg/kg) and were perfused transcardially with phosphate-buffered saline (PBS), followed by 4% paraformaldehyde (PFA, Sigma, 158127MSDS, St. Louis, MO, USA). The brain was removed and post-fixed overnight in 4% PFA at 4 °C, and sectioned coronally at a 40 μm thickness across the whole brain with a cryostat microtome (Thermo Fisher, NX50, Waltham, MA, USA). For anterograde monosynaptic tracing, every sixth section was stained with 40,6-diamidino-2phenylindole (DAPI) and mounted in 70% glycerol. 

For RV tracing samples, one out of three consecutive slices between 1.10 anterior and −0.94 mm of the bregma were selected for following anti-ChAT immunofluorescence staining. For samples in go/no-go tasks, every one out of six consecutive slices between +1.10 mm and −0.70 mm of the bregma were used for anti-ChAT immunofluorescence staining.

Immunohistochemistry staining was performed as a previous study described [4]. Briefly, the sections were washed in PBS (3 times, 5 min each), and blocked for 1 h at 37 °C with 10% bovine serum albumin with 0.3% Triton X-100 in PBS, then incubated with the primary antibody at an appropriate dilution in the blocking buffer overnight at 4 °C. After washing three times (5 min each), the sections were incubated with a second antibody at an appropriate dilution in PBS for 1h at 37 °C, then rinsed in PBS three times (5 min each) and stained with DAPI after. The information on primary and second antibodies were listed as follows: goat anti-ChAT (1:200, ab144P, Millipore, Bedford, MA, USA), rabbit anti-DsRed (1:700, 2250S, Cell Signaling Technology), donkey anti-goat 647 (1:400, 705-605-147, Jackson ImmunoResearch), goat anti-rabbit cy3 (1:400, 111-165-003, Jackson ImmunoResearch), and donkey anti rabbit cy3 (1:400, 711-165-152, Jackson ImmunoResearch) antibodies. Images were acquired by a confocal microscope (Leica, TCS SP8, Buffalo Grove, IL, USA) or a virtual microscopy slide scanning system (Olympus, VS 120, Tokyo, Japan). 

### 4.4. Fiber Implantation and Fiber Photometry

After viral injection, an optical fiber (200 μm o.d., NA: 0.37, 1.5 m long, NEWDOON) was implanted 200 nm above the virus injection site and cemented to the skull with dental acrylic. Meanwhile, a customized aluminum sheet was glued on the skull surface for subsequent head-fixed go/no-go tasks. The mice receiving surgical procedures were kept in separate cages for two weeks. During go/no-go training, fiber photometry recordings were performed and analyzed as previously described [38]. Briefly, the laser beam from a 488 nm laser (OBIS 488LS, Coherent) was reflected by a dichroic mirror (MD498, Thorlabs), focused by an objective lens (10, NA: 0.3; Olympus), and coupled to an optical commutator (Doric Lenses), which was connected to the implanted fiber and coupled by an optical fiber. The power of the laser at the tip of the optical fiber was adjusted to 40–60 μW. The fluorescence emission from GCaMP6s was filtered by a bandpass filter (MF525-39, Thorlabs) and detected by a photomultiplier tube (R3896, Hamamatsu). The current output of the photomultiplier tube was converted into a voltage via an amplifier (C7319, Hamamatsu), then was further filtered through a low-pass filter (35 Hz cutoff; Brownlee, 440). The analogue voltage signal was digitized at 500 Hz and recorded by fiber optic photometric software (Thinkerbiotech). After the experiment, the mice were used for histological studies to verify viral injection and recording sites.

### 4.5. Behavioral Tests

#### 4.5.1. Buried Pellet Tests

A buried pellet test was performed as previously described [39]. Mice were fasted for 24 h before the test. On the first testing day, mice were habituated for 10 min before testing in a cage (27 cm × 15 cm × 14 cm). Then, mice were put into another clean cage containing clean bedding with a food pellet (0.2 g) buried 3 cm deep in a random corner of the cage. The latency was defined as the time from mice entry into the cage to uncovering the pellet within 5 min. On the following day, the procedure was repeated except that the pellet was placed on top of the bedding.

#### 4.5.2. Odor Discrimination Test

Mice were tested in a clean cage (PC, size: 325 × 210 × 185 mm, Feng shi, Suzhou, China) with fresh wood-chip bedding and a metal lid. Before the test, mice were habituated in the testing cage with a Q-tip used as an odor applicator on the metal lid for 3min. During the experiment, the tip of the Q-tip was infiltrated with little odor solution. The sniffing time was defined as the total time when the distance between the mouse’s nose and tip was less than 1 cm that was recorded in each test. Isoamyl acetate and 2-heptanone (0.01%, *v/v*) were used in the odor discrimination test. Each odor exposure consisted of three consequent trials. Every trial was 1 min long and the intertrial interval was 30 s long. The discrimination index was calculated according to the difference between the sniffing times in the first trial of the second odor and the last trial of the first odor. 

#### 4.5.3. Odor-Cued Go/No-Go Discrimination Task

The go/no-go task was performed as previously described [38]. Briefly, mice were deprived of water for 24 h and maintained at 80–85% of their baseline body weights. For go/go training, water-deprived mice were trained to lick water when either of two odors coupled with water delivery emerged within a specific time window. For go/no-go training, isoamyl acetate (0.01%, *v/v*) was set as the rewarding odor (S+) and 2-heptanone (0.01%, *v/v*) was set as the no-rewarding odor (S−). The mice successfully finished the go/go training and were trained to lick water when S+ was present and not to lick when S− was present. The score for S+ presentation with water-licking was marked as a hit, and conversely, without licking as a miss. The score for S− presentation with water-licking was defined as a false alarm (FA), and conversely, without licking as a correct rejection (CR). The hit and CR were considered as correct responses. In all the trainings and tests, the sequence of two odors was randomly selected by the computer. The correct ratio of discrimination behavior was obtained by dividing the sum of the hit and CR by the total number of trials.

### 4.6. Data Analysis

#### 4.6.1. Cell Counting in Tracing Samples

For anterograde monosynaptic tracing samples, cells labeled by both rAAV and HSV were counted as starter cells at the injection site. To count the postsynaptic neurons labeled by HSV in the whole brain, brain sections were segmented and delineated into different brain regions with photoshop based on the Allen Mouse Brain Atlas (2011). (http://www.brain-map.org/, accessed on 17 March 2022). The labeled neurons with mCherry expression were counted with ImageJ (National Institutes of Health, Bethesda, MD, USA). For statistical analysis of the output intensity of the whole brain, all regions were included. To simplify the output intensity across the brain, only regions with >1% proportions were shown in Figure 3. Abbreviations of brain regions were listed in Abbreviations. For RV tracing samples, GFP^+^, mCherry^+^, GFP^+^/ChAT^+^, mCherry^+^/ChAT^+^, and GFP^+^/mCherry^+^/ChAT^+^ neurons were counted, respectively. All values are presented as the mean ± SEM. A *t*-test was used to compare the difference between the two groups. Graphs were drawn with GraphPad Prism 8 (GraphPad Software, San Diego, CA, USA). 

#### 4.6.2. Cell Counting in Go/No-Go Tasks

For samples of both the Casp3 group and control group, ChAT^+^ neurons were counted to evaluate the efficiency of rAAV in killing the ChAT^+^ neurons. 

#### 4.6.3. Analysis of Fiber Photometry Data 

Data were exported as MATLAB .mat files for analysis. In go/no-go, the total 10 s time window in each trial was segmented as 4 s before the onset of odor delivery, 2 s of odor delivery, 2 s of water delivery, and 2 s after water delivery, in order. The values of fluorescence changes (ΔF/F) were calculated by (F − F_0_)/F_0_, where F_0_ is the baseline fluorescence signal before the onset of odor stimulation, which was averaged over a 5 s period of time. They were displayed as heat maps or plot maps via MATLAB software. A *t*-test was used to compare the difference between the two groups. 

#### 4.6.4. Analysis of Behavioral Data

A *t*-test was used to compare the difference between the two groups. Graphs were drawn with GraphPad Prism 8.

## Figures and Tables

**Figure 1 ijms-23-08472-f001:**
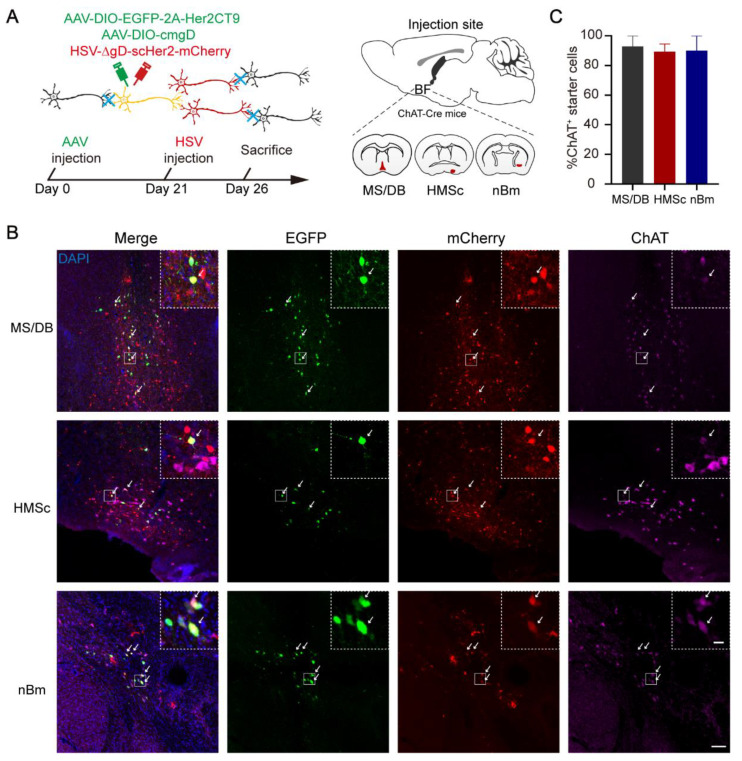
Anterograde monosynaptic tracing. (**A**) Structure (top), time-course of injection (bottom), and injection site (right) of the anterograde monosynaptic virus. (**B**) Representative image of the starter cells in MS/DB (top), HMSc (middle), and nBm (bottom). Enlarged views of starter cells in the top right corner of each image in the first column. Merge image (first column), EGFP (green, second column), mCherry (red, third column), and anti-ChAT immunofluorescence staining (pink, last column) are shown separately. The starter cells labeled by rAAV (EGFP), HSV (mCherry), and anti-ChAT immunofluorescence staining (pink) are indicated by white arrows. Nuclei were detected by DAPI (blue) staining. (**C**) The percent of cholinergic starter cells. MS/DB included MS and vDB; HMSc included HDB, SI, and MCPO. *n* = 4 mice in MS/DB group, *n* = 3 mice in HMSc group, *n* = 3 mice in nBm group. Scale bar, 100 μm; 20 μm for insets.

**Figure 2 ijms-23-08472-f002:**
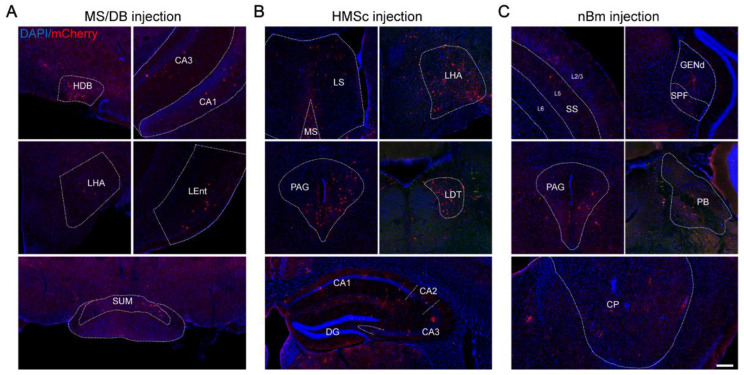
Outputs of the ChAT^+^ cells in different subregions of the BF. (**A**–**C**) Representative brain regions were labeled by anterograde monosynaptic virus injected into three subregions: MS/DB (**A**), HMSc (**B**), and nBm (**C**). Scale bar, 100 μm. DAPI (blue) for the nucleus, HSV (mCherry) for the output neurons.

**Figure 3 ijms-23-08472-f003:**
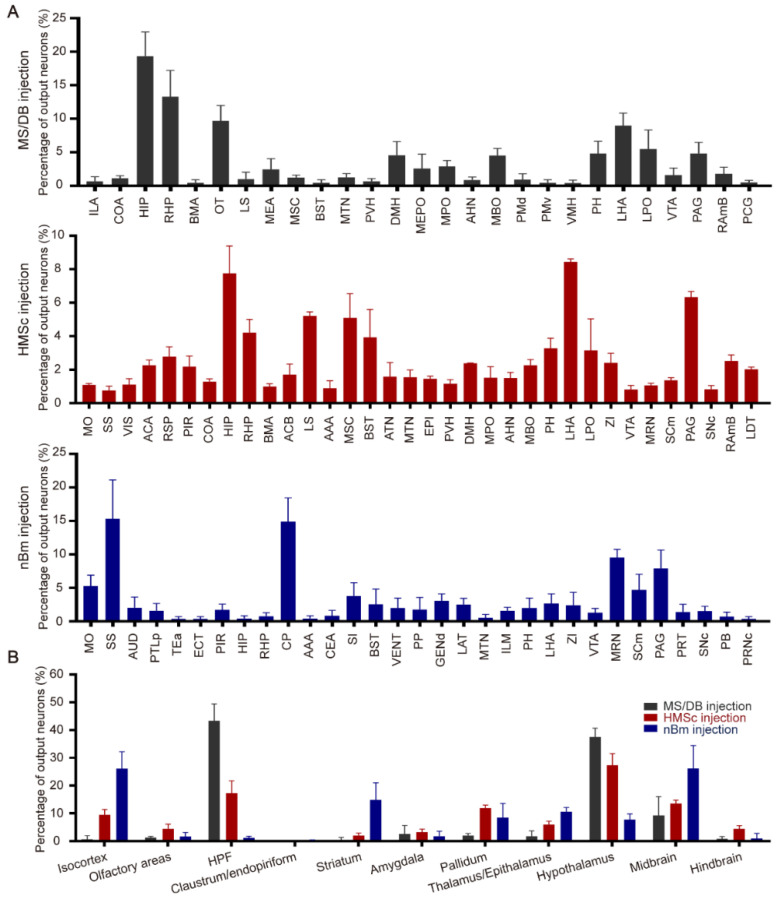
Quantitative analysis of neurons labeled in each brain region across the whole brain. (**A**) Proportion of whole-brain outputs of cholinergic neurons in the different subregions of the basal forebrain, MS/DB (top), HMSc (middle), and nBm (bottom). (**B**) The proportion of given brain regions receiving cholinergic neuronal projections from the three different subregions.

**Figure 4 ijms-23-08472-f004:**
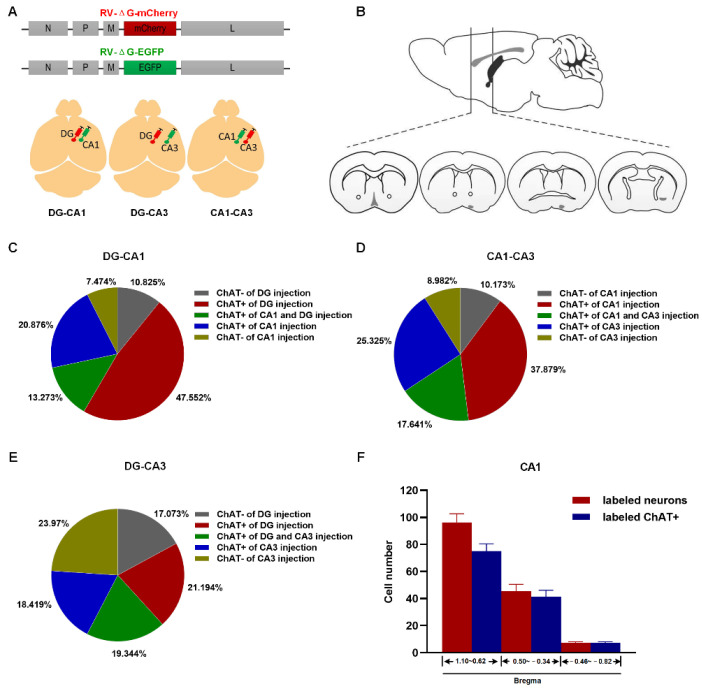
Rabies virus tracing reveals the relationships between BF and dHIP. (**A**) Schematic of double-color rabies virus tracing (top) and injection site (bottom). (**B**) Image acquisition range of neurons labeled by RV. (**C**–**E**) Among all RV-labeled neurons, the proportions of cholinergic to non-cholinergic neurons projecting to distinct subregions of dHIP (CA1, DG, and CA3). (**F**–**H**) The cell numbers of total neurons and the cholinergic neurons are labeled by RV in three subregions of the basal forebrain. *n* = 12, 4 mice in each group.

**Figure 5 ijms-23-08472-f005:**
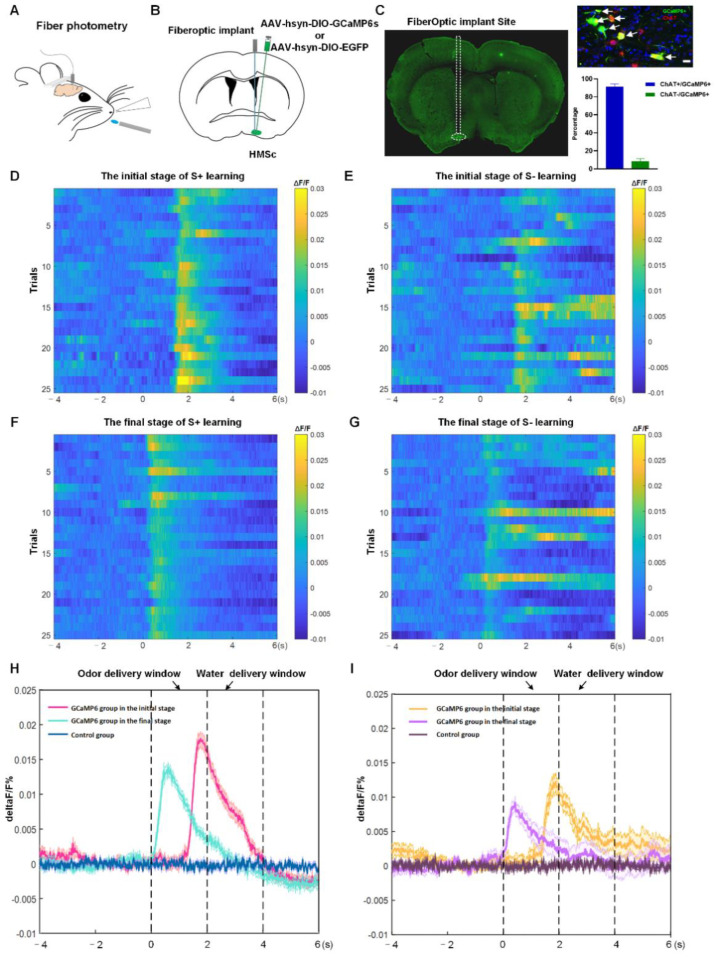
BFCNs in HMSc were activated in the go/no-go olfactory discrimination task. (**A**) Diagram of fiber photometry in head-fixed mice. (**B**) Coronal section schematic showing AAV injection site and fiber optic implanting in the HDB. (**C**) Site of optic fibers (left panel); representative immunofluorescent images of GCaMP6 (green), ChAT (red), and merge (yellow), scale bar: 20 µm (right panel, top); the percent of GCaMP6 expressed in cholinergic neurons in HMSc (right panel, bottom). (**D**,**E**) Heatmap showing responses of the cholinergic neuronal population for S+ and S− signals in the initial stage of go/no-go learning in the GCaMP6 group, respectively. (**F**,**G**) Heatmap showing responses of the cholinergic neuronal population for S+ and S− signals in GCaMP6 group in the final stage of go/no-go learning, respectively. (**H**,**I**) Responses of the cholinergic neuronal population for S+ and S− signals in control and GCaMP6 groups during go/no-go learning. *n* = 12, 5 mice in each group.

**Figure 6 ijms-23-08472-f006:**
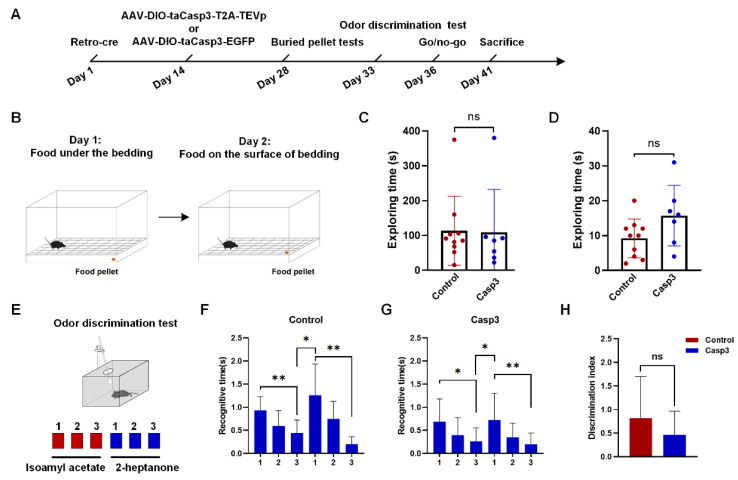
Selective depletion of cholinergic neurons innervating dHIP impaired correct performance of go/no-go olfactory discrimination task. (**A**) Experimental timeline of behavioral tests for selective depletion of cholinergic neurons innervating dHIP affecting olfaction and olfactory learning and memory. (**B**) Diagram of buried pellet tests. Food under the bedding (left), and food on the surface of the bedding (right). (**C**,**D**) Exploring the time of mice in food under the bedding phase and food on the surface of the bedding phase, respectively. (**E**) Diagram of odor discrimination test. (**F**,**G**) Recognition time of mice in control and Casp3 groups for isoamyl acetate and 2-heptanone, respectively. (**H**) Discrimination index of mice in control and Casp3 groups. (**I**,**J**) Schematic and procedure of go/no-go behavioral paradigm. (**K**,**L**) Correctness of response of mice in control and Casp3 groups for S+ and S-, respectively. *n* = 17, 7 mice in the control group and 10 mice in the Casp3 group. (**M**) The number of cholinergic neurons in HMSc between the control and Casp3 groups. A total of 4 mice in each group. * *p* < 0.05, ** *p* < 0.01, *** *p* < 0.001. ns, no significant difference.

**Table 1 ijms-23-08472-t001:** Information of all the viruses listed in the paper.

Virus Name	Company	Cat. No.	Titers
rAAV-hSyn-DIO-EGFP-T2A-Her2CT9-pA	— —	——	1.3 × 10^13^ VG/mL
rAAV-UL26.5p-DIO-cmgD-WPRE-pA	— —	— —	1 × 10^13^ VG/mL
HSV-∆gD-hUbc-mcherry-2A-scHer2::gD-WPRE-pA	— —	— —	1 × 10^8^ pfu/mL
RV-dG-DsRed	BrainVTA	R02001	2 × 10^8^ pfu/mL
RV-dG-GFP	BrainVTA	R02002	2 × 10^8^ pfu/mL
rAAV-EF1α-DIO-GCaMP6s-WPRE-hGH-pA	BrainVTA	PT-0071	2 × 10^12^ VG/mL
rAAV-EF1α-DIO-EGFP-WPRE-hGH-pA	BrainVTA	PT-0795	2 × 10^12^ VG/mL
rAAV-Retro-hSyn-SV40-NLS-Cre-WPRE-hGH-pA	Brain Case	BC-0159	5.93 × 10^12^ VG/mL
rAAV-EF1α-DIO-taCasp3-T2A-TEVp-WPRE-hGH-pA	Brain Case	BC-0130	2.55 × 10^12^ VG/mL
rAAV-EF1α-DIO-taCasp3-EGFP-WPRE-hGH-pA	Brain Case	BC-0846	3.33 × 10^12^ VG/mL

## Data Availability

The authors confirm that the data supporting the findings of this study are available within the article and its Appendix A.

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
