# Peer review of "Basal Forebrain-Dorsal Hippocampus Cholinergic Circuit Regulates Olfactory Associative Learning"

_ijms, 2022, doi:10.3390/ijms23158472_

Round 1

Reviewer 1 Report

This work is an interesting contribution to the knowledge of the relationships between different subpopulations of BFCNs and dHIP, and the possible functions involved. To do so, the authors use a state-of-the-art methodology based on anterograde and retrograde monosynaptic tracing strategies.

The study is comprehensive, covering anatomical, functional and behavioural approaches.

The standard of the manuscript is high. The information is very clear, well organised and informative. The pictures are of very good quality, the writing is correct and the discussion is clear and comprehensive.

The conclusions obtained are convincing and of high interest for understanding the projections of the HMSc complex to the hippocampus.

Certainly, this is a valuable contribution from a group with a long trajectory in the study of the functional role of basal forebrain cholinergic neurons, with highly cited contributions.

My minimal concerns regard the methodology:

- A relevant issue is the method used to investigate the effect of selective depletion on learning and memory. Very appropriately, the authors trained free-moving mice on a go/no-go paradigm. Would it not be interesting to used additionally an object recognition task, a well-established task used to assess learning and memory in rodents? This would be an interesting control method to assess the specificity of the hypothesised role of the HMSc-dHIP cholinergic circuit in regulating olfactory learning.

- How do the authors quantify the efficiency with which rAAV- Retro-hSyn-SV40-NLS-NLS-Cre-WPRE-hGH pA with rAAV-DIO- 244 taCasp3-T2A-TEVp-WPRE-hGH pA selectively kill cholinergic neurons?

Minor issues:

- Acronyms in the abstract should be explained the first time they are employed. For instance:

Line 27-28. Abstract: “heavy projections of cholinergic neurons in HDB/ MCPO/ SI brain regions (HMS complex, HMSc)”

- Figure 1C. Why the Y-axis representing ChAT+ extends to 120 %. It seems to be an erroneous assumption.

Reviewer 2 Report

In this study, the authors examined the innervation targets of basal forebrain cholinergic subpopulations with anterograde monosynaptic and retrograde tracing strategies. Further with fiber photometry calcium activity recording, the authors showed that hippocampus-projecting HMSc cholinergic neurons were activated in odor-cued go/no-go discrimination tasks. Moreover, selective depletion of this group of neurons impaired the performance of odor-cued go/no-go discrimination task. This study thus revealed some interesting cholinergic functions in regulating olfactory associative learning. The study was well organized and presented. I only have one minor comment regarding the interpretation of the result on impaired odor-cued go/no-go discrimination task after cholinergic depletion. It seems that the mice were still able to associate the odors with water delivery in go/go training session, suggesting largely preserved odor associative learning. It was the relearning process from go/go to go/no-go that was significantly impaired, suggesting that the cholinergic signal may play a less role in the original learning process but more involved in further modification of the circuit.
